# Keep the Beam on Track: Stabilizing Reward Trajectories in Guided Decoding

## Abstract

Decoding algorithms play a central role in enhancing the performance of large language models (LLMs) on complex reasoning tasks. A common approach incorporates Process Reward Models (PRMs), which estimate the quality of intermediate reasoning paths and guide the selection of possible continuations. In this setting, our analysis reveals two notable phenomena: reward estimates tend to decline as reasoning progresses, and the reasoning paths exhibit distinct *volatility patterns* across decoding steps depending on whether the paths lead to correct or incorrect final answers. In particular, correct reasoning tends to be associated with stable reward trajectories, while incorrect reasoning often shows high volatility. Motivated by this observation, we propose Volatility-Scaled Guided Decoding (VSGD), a decoding algorithm that prioritizes candidate paths with lower volatility by jointly considering the magnitude of PRM-estimated rewards and the volatility of these rewards across decoding steps. Experiments on datasets including GSM8K and MATH500 indicate that VSGD reduces the volatility of selected reward trajectories and improves the accuracy of the final answer. These findings suggest that considering the temporal dynamics of reward values, in addition to their magnitude, provides a potential direction for enhancing guided decoding in LLMs.

## 1 Introduction

Large language models (LLMs) achieve strong performance across a range of reasoning tasks, including mathematical problem solving (Vaswani et al., 2017; Brown et al., 2020; Wei et al., 2022; Kojima et al., 2022; Touvron et al., 2023; Achiam et al., 2023; Team et al., 2024; Grattafiori et al., 2024). Early studies suggest that carefully designed prompts, such as Chain-of-Thought, can elicit the reasoning capability of LLMs (Wei et al., 2022; Kojima et al., 2022). Beyond prompting, recent work highlights that the decoding algorithm is another crucial factor, as the decoding algorithm governs token selection and influences the reasoning path. (Shi et al., 2024). Among decoding strategies, tree-based methods such as beam search are prominent because the tree-based methods explore multiple candidate reasoning paths in parallel, thereby increasing the likelihood of uncovering promising reasoning paths before generating the final answer (Graves, 2012; Sutskever et al., 2014; Bahdanau et al., 2014; Wu et al., 2016; Kool et al., 2019; Leblond et al., 2021; Meister et al., 2021; Yang et al., 2024). Building on this idea, recent approaches propose guided decoding methods that incorporate additional signals to prioritize promising candidates. These signals range from rule-based constraints (Lu et al., 2021; Welleck et al., 2022) to model-based scoring functions, including reward models that assess the quality of generated sequences (He et al., 2017; Uesato et al., 2022; Krishna et al., 2022; Lightman et al., 2023; Liu et al., 2024; Snell et al., 2024; Wang et al., 2025).

A growing body of work incorporates external reward models into decoding algorithms to provide target-oriented guidance during text generation. Conventional reward models guide generation by evaluating only the final output, which restricts guidance to the end of the generation process. Process Reward Models (PRMs) address this limitation by assigning rewards to intermediate reasoning steps, thereby enabling step-level guidance of incomplete sequences during decoding (Chan et al., 2019; Uesato et al., 2022; Lee et al., 2023; Lightman et al., 2023; Wang et al., 2024; Setlur et al., 2024; Zeng et al., 2025). Guided decoding algorithms leverage these PRM-estimated scores to prioritize incomplete sequences that are more likely to yield high-quality final answers (Chen et al., 2024; Snell et al., 2024; Zhang et al., 2025b; Hu et al., 2025). However, existing approaches primarily emphasize the magnitude of individual reward values, overlooking how these values evolve across successive

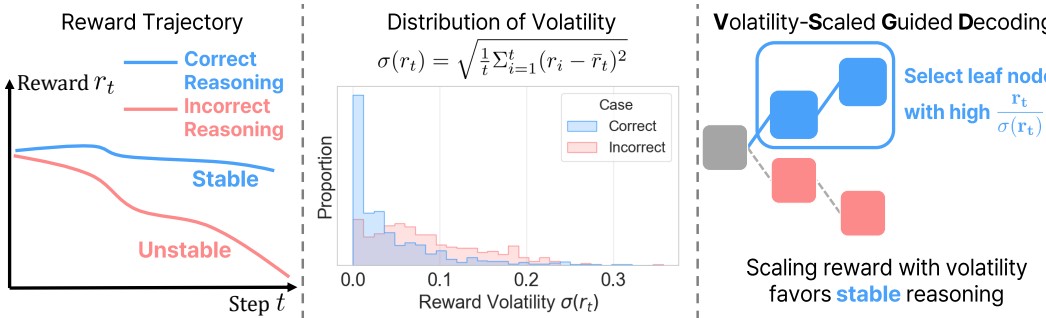

Figure 1: Our empirical observations on the reward trajectory, which motivated our proposed method VSGD. **(Left)** Reward trajectories $r_t$ of process reward model (PRM)–based decoding methods show that correct reasoning paths typically maintain stable and low-volatility rewards, whereas incorrect paths exhibit unstable and high-volatility patterns. **(Middle)** We quantify volatility $\sigma(r_t)$ as a measure of how much rewards fluctuate across decoding steps. Histogram of volatility, computed over 500 samples from MATH500 dataset, shows that correct paths consistently exhibit lower volatility than incorrect paths. **(Right)** Motivated by this finding, we propose Volatility-Scaled Guided Decoding (VSGD), which scales rewards by volatility to favor candidate paths with stable reward trajectories.

reasoning steps. We refer to the sequence of reward scores assigned to each intermediate reasoning step as the *reward trajectory*, denoted by $r_t$. We use the term *volatility* $\sigma(r_t)$ to describe how stable the reward trajectory is across steps. Formally, we measure volatility as the root mean square deviation of the stepwise rewards from their mean. As illustrated in Fig. 1, our empirical analysis suggests an association between volatility and correctness: in observed cases, correct reasoning paths tend to maintain relatively stable reward trajectories, while incorrect reasoning paths show unstable reward trajectories. To obtain these trajectories, we use 500 samples from the MATH500 dataset, following the approach used in Lightman et al. (2023); Wang et al. (2024), which selects candidate continuations based on the minimum PRM-estimated reward across decoding steps. Motivated by this observation, we propose to adjust reward estimates using volatility, so that paths with both high rewards and stable trajectories are given higher priority during decoding.

In this paper, we propose the **V**olatility-**S**caled **G**uided **D**ecoding algorithm (VSGD), a decoding algorithm that integrates the volatility of reward trajectories into the decoding process. We define volatility as the root mean square deviation of reward values across decoding steps, capturing the stability of a reasoning path. Leveraging the volatility measure, VSGD scales reward signals to favor trajectories with more stable rewards, which empirically correlate with correct solutions. Our contributions are as follows:

- We empirically analyze reward trajectories and demonstrate that the volatility, which measures the variability of reward values, exhibits distinct patterns between correct and incorrect reasoning paths.

- Motivated by this observation, we propose Volatility-Scaled Guided Decoding (VSGD), a decoding algorithm that rescales reward values using the volatility of reward trajectory to prioritize stable reasoning paths.

- Experiments on datasets including GSM8K and MATH500, show that VSGD improves reasoning performance by favoring low-volatility reward trajectories. In addition, we show that VSGD maintains superior performance under constrained settings where the number of reasoning steps or the number of complete reasoning paths is limited.

## 2 RELATED WORK

**Guided Decoding Algorithms.** Decoding algorithms are strategies that decide which word or token a language model should generate next, thereby controlling the overall process of text generation. (Graves, 2012; Sutskever et al., 2014; Bahdanau et al., 2014; Wu et al., 2016; Kool et al., 2019; Josifoski et al., 2023). Control over text generation enables additional signals to guide the selection of candidate continuations. The guidance can take the form of rule-based constraints that restrict token selection to satisfy logical requirements (Lu et al., 2021; Welleck et al., 2022), or model-based scoring functions that assign values to incomplete sequences and thereby guide the

search toward preferred continuations (He et al., 2017; Uesato et al., 2022; Krishna et al., 2022; Lightman et al., 2023). Recent studies show that incorporating additional signals into decoding can enhance LLM reasoning by directly affecting the decoding process (Liu et al., 2024; Snell et al., 2024). Tree-based algorithms are particularly suited for guided decoding because tree-based algorithms maintain multiple candidate continuations, allowing additional signals to be applied when ranking or pruning alternatives before the final answer is generated (Yao et al., 2023; Feng et al., 2023; Chen et al., 2024; Yang et al., 2024). Among these, one variant is Monte Carlo Tree Search, which guides decoding by simulating possible continuations (Chaffin et al., 2022; Li et al., 2023a; Feng et al., 2023; Hao et al., 2023). However, the reliance on repeated simulations results in high computational cost (Chaffin et al., 2022; Hao et al., 2023; Liu et al., 2023). By contrast, beam search provides a more favorable trade-off between effectiveness and efficiency, as beam search retains a tractable number of candidate sequences, supporting straightforward integration of external scoring functions (Josifoski et al., 2023; Chen et al., 2024), making it a widely used algorithm for guided decoding.

**Process Reward Models.** While guided decoding typically relies on external signals applied to entire sequences, recent studies incorporate additional signals into decoding algorithms to provide guidance during text generation. Process Reward Models (PRMs) extend this paradigm by assigning rewards to intermediate reasoning steps rather than only to final outputs, thereby enabling step-level evaluation of incomplete sequences (Chan et al., 2019; Uesato et al., 2022; Lee et al., 2023; Lightman et al., 2023; Wang et al., 2024; Setlur et al., 2024; Zeng et al., 2025). Guided decoding algorithms exploit these PRM-estimated scores by aggregating the scores across steps to prioritize sequences that are more likely to generate correct outputs (Chen et al., 2024; Snell et al., 2024; Zhang et al., 2025b; Hu et al., 2025). Nonetheless, most existing approaches emphasize the magnitude of reward values while overlooking the temporal dynamics of reward trajectories.

**Instability of Reward Trajectories.** Instability in reward signals, often called volatility, refers to how much the reward values fluctuate along a reasoning path. In reinforcement learning, Bisi et al. (2019) formalizes volatility as the deviation of cumulative rewards from their average across the entire trajectory. Although this formulation offers a principled measure of instability, it requires access to complete trajectories and, to our knowledge, has not been considered in the context of guided decoding. More recently, Zhang et al. (2025c) introduce *reasoning volatility*, which evaluates the discrepancy between an intermediate reasoning step and the final answer. Concurrently, Zhang et al. (2025a) investigates reward distributions when training PRMs for mathematical reasoning, which reports that reasoning paths leading to correct final answers tend to maintain higher rewards compared to incorrect reasoning paths when tested on the training dataset. In contrast, our study analyzes reward trajectories during the decoding process and introduces *reward volatility* as a metric to quantify the instability of reward trajectories, highlighting the relation between output correctness and reward trajectories during decoding process.

## 3 VSGD: Volatility-Scaled Guided Decoding Algorithm

In this section, we formally describe our proposed algorithm dubbed as VSGD. To begin with, Sec. 3.1 defines the notation and formalizes volatility as a measure of reward instability. Sec. 3.2 presents our observations that reveal the relationship between volatility and the correctness of reasoning paths. Motivated by these observations, in Sec. 3.3 we propose Volatility-Scaled Guided Decoding algorithm (VSGD), which prioritizes reasoning paths characterized by stable reward trajectories.

### 3.1 Notation

Let $x_0$ denote an input prompt, and let $x_t$ represent the tokens generated at decoding step $t$. A reasoning path up to step $t$ is denoted as $\boldsymbol{x}_t = [x_1, \ldots, x_t]$. Let $f$ denote a process reward model (PRM) that maps a reasoning path $\boldsymbol{x}_t$ to a scalar score $f(\boldsymbol{x}_t) \in [0, 1]$, where higher values indicate higher-quality reasoning. The step-level reward at step $t$ is defined as $r_t := f(\boldsymbol{x}_t)$, and the reward trajectory up to step $t$ is denoted as $\boldsymbol{r}_t = [r_1, \ldots, r_t]$. We formalize the *volatility* of the reward trajectory $\boldsymbol{r}_t$ as

$$\sigma(\boldsymbol{r}_t) := \sqrt{\tfrac{1}{t} \sum_{i=1}^{t} (r_i - \bar{r}_t)^2}, \tag{1}$$

where $\bar{r}_t = \frac{1}{t} \sum_{i=1}^{t} r_i$ is the reward averaged over the reasoning step $t$. This formulation corresponds to the standard deviation of a time series, a measure widely used across domains including

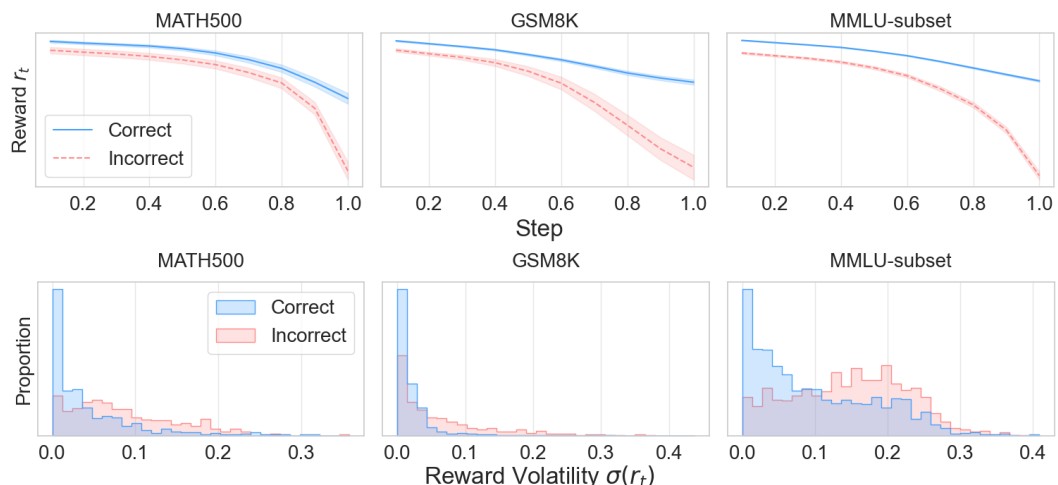

Figure 2: Dynamics of correct and incorrect reasoning paths under PRM-guided decoding as established in a prior work (Zeng et al., 2025) on MATH500, GSM8K and MMLU-subset. Correct paths are shown in blue and incorrect paths in red. (**Top**) Reward trajectories with standard error shading show that correct paths maintain higher rewards throughout decoding, while incorrect paths decline. (**Bottom**) Volatility histograms (Eq. 1) indicate that correct paths generally exhibit lower volatility than incorrect paths across benchmarks.

finance (Markowitz, 1952; Daly, 2008; 2011; Alan & Kyre, 2019), physiology (Brockmann & Hunt, 2023), and reinforcement learning (Bisi et al., 2019). In this work, we extend this notion to quantify the variability of PRM-estimated rewards across decoding steps. Notably, our formulation differs from that of Zhang et al. (2025c), who define *reasoning volatility* as the semantic discrepancy between an intermediate reasoning step and the final answer. In contrast, we define *reward volatility* as the variability of PRM-estimated rewards within the decoding process.

### 3.2 OBSERVATION ON REWARD TRAJECTORY

Here we present empirical results showing how conventional PRM guidance shapes decoding dynamics. Specifically, we compare the reward trajectories of two cases: (i) correct reasoning paths that lead to a correct answer, and (ii) incorrect paths that end with a wrong answer. We evaluate on three datasets: two mathematical reasoning benchmarks – GSM8K (Cobbe et al., 2021) and MATH500 (Lightman et al., 2023) – and a subset of MMLU dataset (Hendrycks et al., 2021). For the latter dataset, we focus on six categories (computer science, chemistry, econometrics, formal logic, philosophy, and virology) in MMLU, and refer to it as the "MMLU-subset" throughout the paper. To analyze reward trajectories, we apply a PRM-based guided decoding approach, following standard configurations from prior work (Zeng et al., 2025). Since the number of steps in reward trajectories varies across samples, we rescale each one to the unit interval [0,1]. We then use linear interpolation to map rewards onto this common scale, allowing us to compare trajectories point by point.

Fig. 2 shows the reward trajectory $r_t$ and its volatility $\sigma(r_t)$ defined in Eq. 1, to quantify the instability of reward trajectories. As shown in the top row of Fig. 2, reasoning paths leading to correct final answers consistently maintain higher rewards, whereas reasoning paths ending in incorrect answers exhibit declines, indicating systematic differences in reward trajectories by correctness. The bottom row of Fig. 2 shows that correct reasoning paths generally have lower volatility than incorrect paths[1]. For each dataset, we assess statistical significance using the Mann–Whitney U test (Mann & Whitney, 1947), which is a non-parametric method for testing whether two distributions differ. For GSM8K, MATH500 and all categories of MMLU-subset, the differences are significant ($p < 0.05$), confirming

---

[1]Concurrently, Zhang et al. (2025a) observe that when training PRMs, correct reasoning paths in the training dataset exhibit higher reward values than incorrect reasoning paths. In contrast, we demonstrate that correct reasoning paths not only maintain higher reward values during decoding but also exhibit distinct *volatility* patterns, which we quantify explicitly.

---

**Algorithm 1** `RescaleWithVolatility`

---

**Input:** Reward trajectory $\boldsymbol{r}_t = [r_1, \ldots, r_t]$, stability constant $\epsilon$
**Output:** Rescaled reward trajectory $\tilde{\boldsymbol{r}}_t = [\tilde{r}_1, \cdots, \tilde{r}_t]$
1: $\bar{r} \leftarrow \frac{1}{t} \sum_{i=1}^{t} r_i$
2: $\sigma \leftarrow \sqrt{\frac{1}{t} \sum_{i=1}^{t} (r_i - \bar{r})^2}$
3: $\tilde{\boldsymbol{r}}_t \leftarrow [\, r_i/(\sigma + \epsilon) \,]_{i=1}^{t}$
4: **return** $\tilde{\boldsymbol{r}}_t$

---

**Algorithm 2** Volatility-Scaled Guided Decoding (VSGD)

---

**Input:** PRM $f$, prompt $\boldsymbol{x}_0$, expansion width $M$, beam size $N$, maximum step $T$, maximum number of complete reasoning paths $L$, stability constant $\epsilon$, reward trajectory aggregation function `Agg`
**Output:** Complete reasoning path list $\mathcal{X}_{\text{comp}}$
**Constraint:** $N$ is divisible by $M$
1: $\mathcal{X}_0 \leftarrow [\, (\boldsymbol{x}_0, [\,]) \mid i = 1, \ldots, N/M \,]$        ▷ List of (reasoning path, reward trajectory) pairs
2: $\mathcal{X}_{\text{comp}} \leftarrow [\,]$        ▷ List of complete reasoning paths
3: **for** $t = 1$ to $T$ **do**
4:      $\mathcal{C}_t, \mathcal{S}_t \leftarrow [\,], [\,]$        ▷ Lists of incomplete reasoning paths and their aggregated scores
5:      **for all** $(\boldsymbol{x}_{t-1}, \boldsymbol{r}_{t-1}) \in \mathcal{X}_{t-1}$ **do**
6:          **for all** $\boldsymbol{x}_t \in \text{Expand}(\boldsymbol{x}_{t-1}, M)$ **do**
7:              **if** $\boldsymbol{x}_t$ ends with `<eos>` **then**
8:                  $\mathcal{X}_{\text{comp}} \leftarrow \mathcal{X}_{\text{comp}}.\text{append}(\boldsymbol{x}_t)$        ▷ Store the complete reasoning path
9:                  **if** $|\mathcal{X}_{\text{comp}}| \geq L$ **then**
10:                      **return** $\mathcal{X}_{\text{comp}}$
11:                  **end if**
12:                  **continue**
13:              **end if**
14:              $\boldsymbol{r}_t \leftarrow \boldsymbol{r}_{t-1}.\text{append}(f(\boldsymbol{x}_t))$
15:              $\tilde{\boldsymbol{r}}_t \leftarrow \text{RescaleWithVolatility}(\boldsymbol{r}_t, \epsilon)$
16:              $\mathcal{C}_t \leftarrow \mathcal{C}_t.\text{append}((\boldsymbol{x}_t, \boldsymbol{r}_t))$
17:              $\mathcal{S}_t \leftarrow \mathcal{S}_t.\text{append}(\text{Agg}(\tilde{\boldsymbol{r}}_t))$
18:          **end for**
19:      **end for**
20:      best_beam_indices $\leftarrow \text{Argsort}(\mathcal{S}_t, \frac{N}{M})$
21:      $\mathcal{X}_t \leftarrow \mathcal{C}_t[\text{best\_beam\_indices}]$
22: **end for**
23: **return** $\mathcal{X}_{\text{comp}}$

---

that volatility distributions for correct and incorrect reasoning paths differ significantly; the $p$-value for each dataset is given in Table 4 in Appendix.

### 3.3 Decoding Algorithm

We present the Volatility-Scaled Guided Decoding (VSGD) algorithm, which incorporates reward volatility into a step-level beam search framework to prioritize reasoning paths exhibiting stable reward trajectories. In this framework, beam search maintains a fixed number of candidate sequences at each step, and VSGD ranks candidate sequences using reward values scaled by their volatility. To be specific, Algorithm 1 describes the scaling procedure used in VSGD algorithm. Given a reward trajectory $\boldsymbol{r}_t$, we compute its volatility $\sigma(\boldsymbol{r}_t)$ as in Eq. 1. Each reward value is then divided by $(\sigma(\boldsymbol{r}_t) + \epsilon)$, where $\epsilon = 10^{-6}$ prevents division by zero. This rescaling penalizes trajectories with high volatility by attenuating their reward values, thereby highlighting trajectories with more stable reward patterns.

Algorithm 2 outlines the overall decoding process of VSGD. At each step $t$, for each incomplete sequence $\boldsymbol{x}_{t-1}$, the function $\text{Expand}(\boldsymbol{x}_{t-1}, M)$ generates $M$ candidate continuations by using $\boldsymbol{x}_{t-1}$ as the prefix. The PRM then assigns a step-level reward $r_t = f(\boldsymbol{x}_t)$ for each sequence

Table 1: Comparison of VSGD with baseline decoding algorithms on GSM8K, MATH500, and an MMLU-subset. The MMLU-subset comprises six categories: computer science (CS), chemistry (Chem), econometrics (Econ), formal logic (Logic), philosophy, and virology. Final answers are selected with Best-of-N (BoN) and Weighted Majority Voting (WMV); accuracy is reported with standard deviation as subscripts. Best and second-best results per dataset are shown in bold and underlined, and $\Delta$ indicates the accuracy difference between VSGD and the strongest baseline for each dataset. On average, VSGD improves over the strongest baseline by 1.1 points under BoN and 1.4 points under WMV.

| Selection | BoN | | | | | WMV | | | | |
|---|---|---|---|---|---|---|---|---|---|---|
| Dataset | GD-Sum | GD-Last | GD-Min | VSGD | $\Delta$ | GD-Sum | GD-Last | GD-Min | VSGD | $\Delta$ |
| MATH500 | $48.9_{0.3}$ | $49.5_{0.9}$ | $\underline{50.2_{0.8}}$ | $\mathbf{50.8_{0.9}}$ | $+0.6$ | $49.8_{0.5}$ | $49.5_{0.7}$ | $\underline{50.1_{0.8}}$ | $\mathbf{50.8_{0.8}}$ | $+0.7$ |
| GSM8K | $86.3_{0.4}$ | $\underline{87.9_{0.2}}$ | $\mathbf{88.1_{0.2}}$ | $\underline{87.9_{0.5}}$ | $-0.2$ | $87.2_{0.4}$ | $\underline{88.2_{0.2}}$ | $87.7_{0.2}$ | $\mathbf{88.3_{0.5}}$ | $+0.1$ |
| CS | $50.3_{1.8}$ | $50.7_{0.3}$ | $\underline{51.7_{0.3}}$ | $\mathbf{52.3_{2.7}}$ | $+0.6$ | $\mathbf{56.0_{0.6}}$ | $53.3_{0.9}$ | $53.7_{0.3}$ | $\underline{55.3_{2.3}}$ | $-0.7$ |
| Chem | $40.3_{2.8}$ | $\underline{44.7_{0.9}}$ | $43.0_{1.2}$ | $\mathbf{46.3_{1.5}}$ | $+1.6$ | $\underline{44.3_{1.5}}$ | $43.0_{0.6}$ | $43.3_{1.2}$ | $\mathbf{45.3_{1.3}}$ | $+1.0$ |
| Econ | $51.2_{0.6}$ | $\underline{52.9_{1.2}}$ | $53.2_{1.5}$ | $\mathbf{53.8_{2.5}}$ | $+0.6$ | $50.9_{1.3}$ | $51.5_{0.8}$ | $\underline{52.9_{0.6}}$ | $\mathbf{54.4_{1.8}}$ | $+1.5$ |
| Logic | $45.5_{1.4}$ | $\underline{47.4_{0.3}}$ | $46.6_{1.1}$ | $\mathbf{48.4_{0.9}}$ | $+1.0$ | $46.3_{2.5}$ | $46.6_{1.9}$ | $\underline{49.5_{0.7}}$ | $\mathbf{49.7_{0.5}}$ | $+0.2$ |
| Philosophy | $60.2_{1.1}$ | $\underline{62.0_{1.1}}$ | $62.0_{0.6}$ | $\mathbf{63.3_{1.9}}$ | $+1.3$ | $61.8_{1.1}$ | $\underline{62.3_{1.1}}$ | $60.6_{0.7}$ | $\mathbf{63.6_{2.0}}$ | $+1.3$ |
| Virology | $\mathbf{47.0_{0.6}}$ | $45.8_{1.5}$ | $46.4_{0.9}$ | $47.0_{0.3}$ | $0.0$ | $\underline{46.8_{1.0}}$ | $46.2_{0.9}$ | $46.4_{1.0}$ | $\mathbf{47.4_{0.4}}$ | $+0.6$ |
| Average | $53.7_{0.6}$ | $55.1_{0.3}$ | $\underline{55.1_{0.1}}$ | $\mathbf{56.2_{0.7}}$ | $+1.1$ | $55.4_{0.5}$ | $55.1_{0.2}$ | $\underline{55.5_{0.3}}$ | $\mathbf{56.9_{0.7}}$ | $+1.4$ |

$x_t$. We append $r_t$ to $r_{t-1}$, forming $r_t$, and compute the rescaled reward trajectory $\tilde{r}_t$ using `RescaleWithVolatility` defined in Algorithm 1. We will use this rescaled reward trajectory $\tilde{r}_t$ for ranking the reasoning path candidates.

We define the *candidate list* $\mathcal{C}_t$ which stores all candidates of reasoning path $x_t$ as well as its reward sequence $r_t$. Let Agg be an aggregation function which return a scalar value for a given input vector, *e.g.,* the minimum value within the input vector. We define the *aggregated score list* $\mathcal{S}_t$ which stores the scalar score obtained by applying the aggregation function Agg to $\tilde{r}_t$, where the choice of aggregation depends on the decoding algorithm. Given the aggregated score list $S_t$, we sort them to choose $N/M$ good candidates. Let $\mathrm{Argsort}(\mathcal{S}_t, k)$ be the operation that returns the indices of the top-$k$ elements in $\mathcal{S}_t$, sorted in descending order of their scores. Then, the beam set $\mathcal{X}_t$ is updated by applying $\mathrm{Argsort}(\mathcal{S}_t, \frac{N}{M})$ to identify the indices of the $\frac{N}{M}$ highest scores in $\mathcal{S}_t$, and subsequently selecting the corresponding tuples $(x_t, r_t)$ from $\mathcal{C}_t$.

When a reasoning sequence $x_t$ in the updated beam set $\mathcal{X}_t$ reaches the end-of-sequence token <eos>, we call such sequence is *complete*, and store the complete sequences $x_t$ in the list of complete reasoning paths $\mathcal{X}_{\mathrm{comp}}$. The decoding process terminates when either the maximum reasoning step $T$ is reached, or all sequences end, or the maximum number of completed reasoning paths $L$ is obtained, after which the algorithm returns the list of complete reasoning paths $\mathcal{X}_{\mathrm{comp}}$.

## 4 EXPERIMENTAL RESULT

We evaluate our proposed VSGD and conventional PRM-based decoding algorithms on two mathematical reasoning benchmarks, GSM8K and MATH500, as well as on the MMLU-subset. Sec. 4.1 reports the main results, showing that VSGD generally achieves higher accuracy on most benchmark datasets we tested. In addition, VSGD makes efficient use of the computational budget, consistently outperforming baseline methods across different limits on decoding steps $T$ and the number of complete reasoning paths $L$. Sec. 4.2 further analyzes the factors contributing to the performance of VSGD, with particular focus on the reduced volatility of reward trajectories selected by VSGD compared to an existing decoding algorithm. We also examine how VSGD improves reasoning quality, finding that VSGD tends to mitigate the issue of incomplete reasoning. Our implementation is publicly available at `https://anonymous.4open.science/r/Volatility-Scaled-Guided-Decoding-B26C/README.md`.

**Datasets.** We evaluate VSGD on three datasets: two mathematical reasoning benchmarks – GSM8K (Cobbe et al., 2021) and MATH500 (Lightman et al., 2023)– as well as MMLU-subset. Here, MMLU-subset is a subset of MMLU dataset (Hendrycks et al., 2021), consisting of six subject areas: computer science, chemistry, econometrics, formal logic, philosophy, and virology. For all

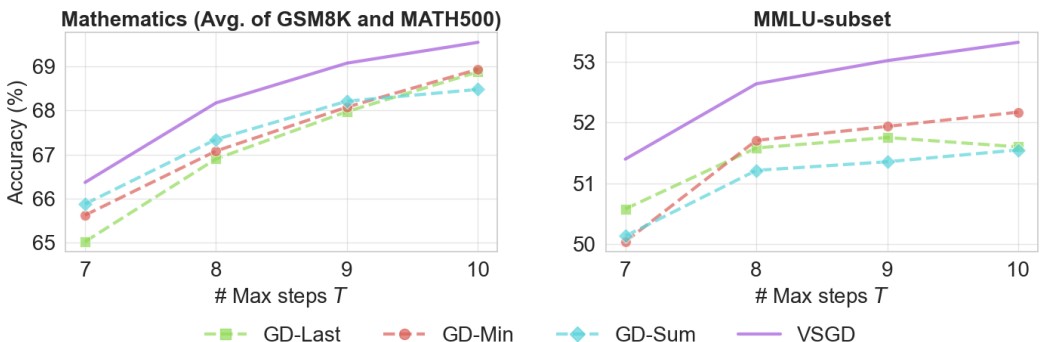

Figure 3: Accuracy of decoding algorithms under varying maximum number of reasoning step $T$ in Algorithm 2. Results are reported on (**Left**) mathematical reasoning datasets (GSM8K and MATH500) and (**Right**) the MMLU-subset. The $x$-axis indicates the maximum number of reasoning steps $T$, and the $y$-axis denotes the accuracy of the final answer. For each $T$, only responses whose reasoning paths do not exceed $T$ steps are retained when computing accuracy. Accuracy generally improves as $T$ increases, and VSGD consistently outperforms the baselines, suggesting that VSGD can effectively utilize longer reasoning paths.

benchmark datasets, we report the average accuracy over three random seeds. The number of test queries for each dataset is listed in Table 3 in Appendix.

**Models.**    We adopt LLAMA-3.1-8B-INSTRUCT (Llama-8B) (Grattafiori et al., 2024) as the base LLM and VersaPRM (Zeng et al., 2025) as the base PRM. It is notable that VersaPRM is trained on datasets covering subjects that include domains in the MMLU-subset and mathematical reasoning datasets. Unless otherwise specified, Llama-8B is used as the default LLM and VersaPRM as the default PRM throughout the experiments. All decoding methods are configured with the same LLM and PRM settings to ensure a fair comparison.

**Baselines.**    We compare VSGD against baseline decoding algorithms that rank or expand incomplete sequences using step-level rewards estimated by a PRM. GD-Sum selects candidate continuations based on the total accumulated PRM rewards across decoding steps where we use PRMs as the state evaluator (Feng et al., 2023; Yao et al., 2023). GD-Min evaluates candidates by the minimum PRM-estimated reward across decoding steps (Lightman et al., 2023; Wang et al., 2024). GD-Last considers only the PRM-estimated reward at the final decoding step (Snell et al., 2024).

Our implementation builds on the decoding algorithm of Snell et al. (2024) to incorporate VSGD. We set the total beam size to $N = 8$ and the expansion width to $M = 2$. At each decoding step, the top $N/M = 4$ incomplete sequences with the highest aggregated scores, computed from rescaled reward trajectories $\tilde{r}_t$, are retained. Each retained sequence is then expanded into $M = 2$ candidates, yielding $N = 8$ candidate continuations for the next step. This procedure is repeated until a complete response is generated, with at most $T = 10$ reasoning steps and up to $L = 32$ complete reasoning paths, which together constrain the computational budget. For the reward trajectory aggregation function Agg, we follow prior work which use the minimum reward value of reward trajectories $r_t$, $i.e.,$ $\min_{i \in [t]} r_i$ (Sun et al., 2024; Wang et al., 2024; Zeng et al., 2025). For the final answer selection, we consider two strategies. The Best-of-$N$ (BoN) strategy selects the candidate answer with the highest aggregated score, while the Weighted Majority Voting (WMV) strategy determines the final answer through voting, using the aggregated scores as weights (Li et al., 2023b).

### 4.1 DOES VSGD OUTPERFORM EXISTING GUIDED DECODING METHODS?

**Yes, VSGD generally outperforms existing guided algorithms.**    Table 1 presents the performances of VSGD and baseline decoding methods, using Llama-3 as the base model and VersaPRM as the process reward model. Results are reported under two aggregation strategies (BoN and WMV), each averaged over three random seeds. Overall, VSGD achieves the highest average accuracy under both aggregation strategies, yielding relative improvements of 1.1 points over the strongest baseline in BoN and 1.4 points in WMV. At the dataset level, VSGD delivers overall gains: in Chemistry, it

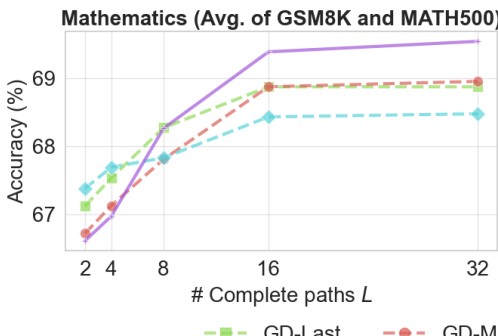 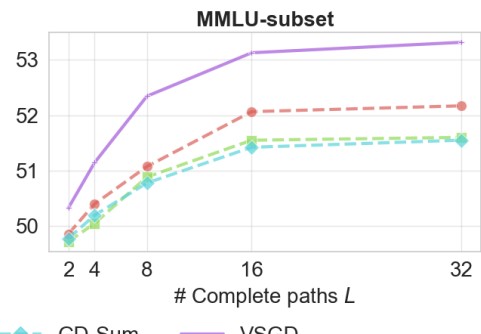

Figure 4: Accuracy of decoding algorithms with a limited number of reasoning paths $L$ in Algorithm 2. We report the average accuracy on (**Left**) mathematical reasoning datasets, which consist of GSM8K and MATH500 and (**Right**) the MMLU-subset. The $x$-axis indicates the number of reasoning paths $L$, and the $y$-axis denotes accuracy of the final answer. For each $L$, we consider the first $L$ reasoning paths that reach termination earliest during decoding. Across decoding algorithms, accuracy improves with larger $L$, reflecting the benefit of utilizing additional reasoning paths. This trend is pronounced for VSGD, which more effectively use additional reasoning paths compared to the baselines.

achieves a 1.6 point improvement under BoN; in Economics, a 1.5 point improvement under WMV; and in Philosophy, a 1.3 point improvement under both BoN and WMV. These gains are realized without substantial degradation in domains where baselines already achieve high accuracy, where performance of VSGD remains competitive despite minor drops. These results suggest that VSGD obtains performance improvements across both mathematical reasoning tasks and MMLU-subset.

**Yes, VSGD effectively exploits reasoning steps.** Fig. 3 compares the performances of VSGD and baseline decoding methods, by varying the maximum reasoning step $T$ in Algorithm 2. Recall that all experiments are conducted with the default budget $T = 10$. To analyze performance under smaller budgets ($T < 10$), we exclude reasoning paths whose length exceeds $T$, and then compute accuracy on the remaining reasoning paths. Across all decoding algorithms, accuracy increases as $T$ grows, highlighting the benefit of longer reasoning trajectories. Moreover, VSGD consistently surpasses the baselines, demonstrating stronger capability in leveraging reasoning steps to reach correct solutions.

**Yes, VSGD achieves higher accuracy with additional reasoning paths.** We consider a scenario where only the first $L$ complete reasoning paths are used for evaluation. By default, we set $L = 32$, and here we investigate how performance varies as $L$ increases. Fig. 4 reports the accuracy of decoding algorithms as $L$ varies in $\{2, 4, 8, 16, 32\}$. Our experimental results show that, across decoding algorithms, accuracy generally increases with larger $L$, as more reasoning pahts become available. Moreover, VSGD generally achieves higher accuracy than the baselines, indicating the effectiveness of VSGD in using additional reasoning paths compared to existing baseline methods.

### 4.2 What effects does VSGD exhibit in decoding?

**VSGD prioritizes stable reward trajectories.** Table 2 reports a comparative analysis of volatility between VSGD and GD-Min. We use GD-Min as the baseline because it selects trajectories solely based on PRM reward scores without considering reward volatility, whereas VSGD incorporates reward volatility by weighting trajectories with more stable reward signals. The $\Delta$ column in Table 2 reports the relative reduction (%) of VSGD compared to GD-Min. Compared to GD-Min, VSGD shows lower volatility $\sigma(\boldsymbol{r}_t)$ defined in Eq. 1, measured across the decoding steps. For example, volatility is reduced by $-9.48\%$ on the CS category and by $-8.92\%$ on Philosophy, with an average reduction

Table 2: Comparison between VSGD and GD-Min, in terms of volatility of the trajectory.

| Dataset | GD-Min | VSGD | $\Delta$ (%) |
|---|---|---|---|
| MATH500 | 0.0587 | 0.0537 | -8.49 |
| GSM8K | 0.0267 | 0.0246 | -7.82 |
| CS | 0.1380 | 0.1249 | -9.48 |
| Chem | 0.1041 | 0.1081 | 3.85 |
| Econ | 0.1543 | 0.1438 | -6.75 |
| Logic | 0.1345 | 0.1303 | -3.10 |
| Philosophy | 0.1100 | 0.1002 | -8.92 |
| Virology | 0.1216 | 0.1180 | -2.96 |
| Average | 0.1065 | 0.1010 | -5.13 |

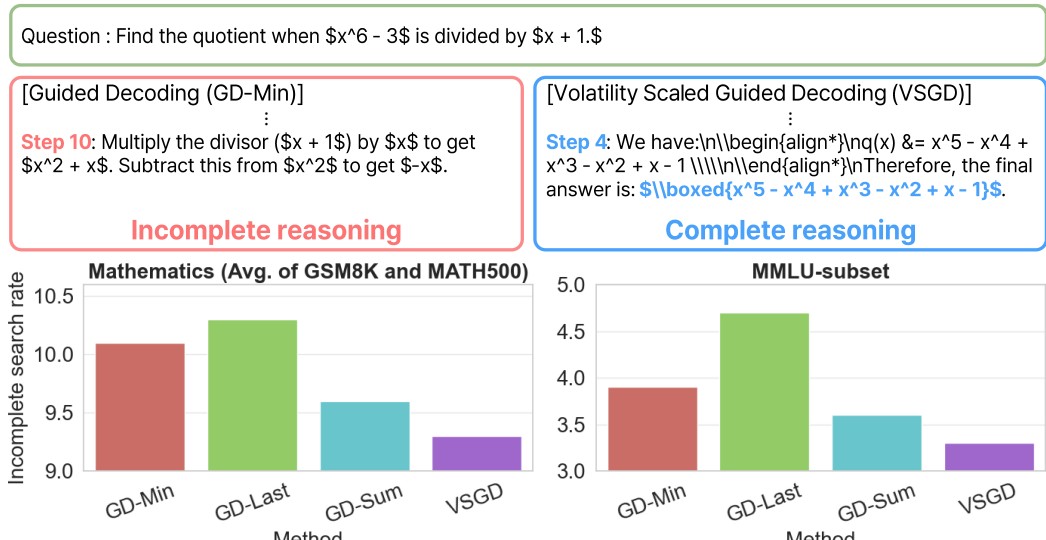

Figure 5: Qualitative and quantitative results showing that VSGD mitigates incomplete reasoning. (**Top**) Example reasoning paths from the MATH500 dataset: the baseline method `GD-Min` reaches the maximum step limit $T = 10$ without generating the final answer, while VSGD generates a complete reasoning path with the final answer. (**Bottom**) Empirical comparison of incomplete search rates on the MMLU-subset and mathematics benchmarks which is comprised of GSM8K and MATH500. For each dataset and decoding algorithm, we compute the proportion of cases that reach the maximum step limit $T$ without generating the final answer, referred to as the incomplete search rate. We report averages across datasets for each decoding algorithm. Our experimental results show that VSGD consistently yields lower incomplete search rates than baseline methods.

of $-5.13\%$ across all datasets. The reduction across benchmarks is statistically significant under the Wilcoxon signed-rank test (Wilcoxon, 1945) yielding $p = 0.0098$, which is appropriate given that both methods are evaluated on the same datasets. These results demonstrate that VSGD effectively guides decoding toward reasoning paths whose reward trajectories are less volatile compared to `GD-Min`.

**VSGD improves accuracy by mitigating failed searches and incomplete reasoning.** One key advantage of VSGD over baseline methods is its ability to reduce cases where the reasoning process remains unfinished. Fig. 5 provides both qualitative and quantitative evidences. The top panel of Fig. 5 provides a qualitative example from the MATH500 dataset: the baseline method `GD-Min` reaches the maximum step limit $T = 10$ without producing a final answer, whereas VSGD completes the reasoning in only four steps and outputs the correct solution. The bottom panel of Fig. 5 reports quantitative results on the MMLU-subset and mathematics (GSM8K and MATH500) benchmarks. We measure the proportion of test cases where decoding terminates without producing the final answer, which we define as the *incomplete search rate*. Across all datasets and methods, VSGD consistently achieves substantially lower incomplete search rates than baselines, demonstrating its effectiveness in generating complete reasoning trajectories.

## 5 CONCLUSION

We introduced Volatility-Scaled Guided Decoding (VSGD), a decoding algorithm that leverages volatility in reward trajectories within a step-level beam search framework. Our analysis revealed that correct reasoning paths consistently exhibit lower volatility than incorrect ones, motivating the use of volatility as a stability criterion in decoding. Experiments on GSM8K, MATH500, and the MMLU-subset show that VSGD outperforms strong baselines in accuracy while using computational resources more efficiently. By reducing volatility, VSGD improves the rate of successful searches and thus explains its performance gains. More broadly, these results highlight that modeling the temporal dynamics of reward trajectories offers a principled foundation for guided decoding in LLMs.

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

Table 3: Number of evaluation queries used per dataset.

| Dataset | # Test Queries |
|---|---|
| *Task category: Mathematical reasoning* | |
| GSM8K | 1319 |
| MATH500 | 500 |
| *Task category: MMLU-subset knowledge* | |
| Computer Science | 100 |
| Chemistry | 100 |
| Econometrics | 114 |
| Formal Logic | 126 |
| Philosophy | 311 |
| Virology | 166 |

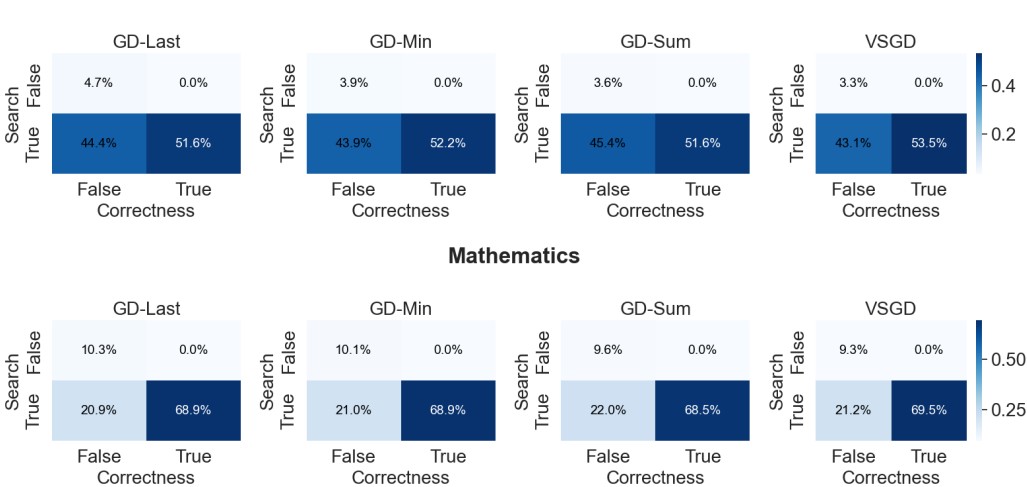

Figure 6: Distribution of outcomes across guidance methods, categorized by search completion (y-axis) and answer correctness (x-axis). Each heatmap illustrates the proportion of examples in four categories defined by whether the decoding algorithm completes a search and whether the final answer is correct. For instance, the *Search=True, Correct=True* cell corresponds to accurate responses, while *Search=True, Correct=False* denotes cases where the search completes but yields an incorrect answer. Across both the (**Top**) MMLU-subset and (**Bottom**) mathematics datasets, VSGD generally results in fewer search failures and fewer incorrect answers upon completion compared to baseline methods, producing a higher proportion of accurate responses. These results demonstrate that VSGD improves reasoning performance by enhancing both search completion and answer correctness.

## A EXPERIMENTS DETAILS

We evaluate our models on a subset of the MMLU benchmark covering six domains. Among these, Chemistry and Computer Science are selected from the college-level categories of MMLU, while the remaining subjects represent diverse knowledge domains. The number of test samples for each dataset is summarized in Table 3.

To ensure efficient inference, we apply 4-bit quantization to all models and enable FlashAttention in every experiment. This configuration reduces memory consumption and accelerates attention computation without compromising accuracy. All experiments are conducted on NVIDIA RTX 4090 GPUs.

# B COMPREHENSIVE RESULTS

Fig. 6 presents the distribution of outcomes by jointly considering search completion and answer correctness. We evaluate decoding algorithms by partitioning examples into four categories: (i) complete search with a correct answer, (ii) complete search with an incorrect answer, (iii) failed search with a correct answer, which is essentially absent, and (iv) failed search with an incorrect answer. This decomposition highlights how decoding algorithms influence the reasoning performance of LLMs. Across both the MMLU-subset and mathematical reasoning datasets including GSM8K and MATH500, VSGD consistently shows lower rates of search failure compared to baseline methods. Moreover, within the subset of successful searches, VSGD generally produces a smaller fraction of incorrect answers, suggesting that the guidance of VSGD is associated with both higher completion rates and improved output quality. As a result, VSGD yields a higher proportion of accurate responses and a lower proportion of failed or misleading trajectories relative to the baselines. These findings suggest that the performance gains of VSGD can be attributed to two complementary factors: reducing the frequency of failed searches and improving correctness within successful searches.

Table 4: Mann–Whitney U test $p$-values showing significant differences in volatility distributions between correct and incorrect reasoning paths across domains.

| Dataset | p-value |
|---|---|
| GSM8K | $1.1 \times 10^{-36}$ |
| MATH500 | $7.4 \times 10^{-49}$ |
| Computer Science | $2.2 \times 10^{-3}$ |
| Chemistry | $1.4 \times 10^{-5}$ |
| Econometrics | $2.6 \times 10^{-7}$ |
| Formal Logic | $1.2 \times 10^{-5}$ |
| Philosophy | $4.8 \times 10^{-52}$ |
| Virology | $1.9 \times 10^{-8}$ |

Table 4 reports the $p$-values from the Mann–Whitney U test across mathematical reasoning benchmarks (GSM8K and MATH500) as well as the MMLU subset, which includes Chemistry, Computer Science, Philosophy, Econometrics, Formal Logic, and Virology. In all cases, the $p$-values are well below the $0.05$ threshold, indicating that the volatility distributions of correct and incorrect reasoning paths differ significantly across domains, thereby reinforcing the patterns observed in Fig. 2.

