# OpenReview forum: "Keep the Beam on Track: Stabilizing Reward Trajectories in Guided Decoding"
_ICLR.cc/2026/Conference — Submitted to ICLR 2026_

### Official Review · Reviewer_ZvW3 · 2025-10-28

**Soundness:** 3
**Presentation:** 2
**Contribution:** 2
**Rating:** 4
**Confidence:** 3

**Summary:**

This paper proposes Volatility-Scaled Guided Decoding (VSGD), a decoding algorithm for large language models that stabilizes Process Reward Model (PRM) guidance by accounting for the volatility of reward trajectories during reasoning. The authors observe that correct reasoning paths exhibit lower reward volatility than incorrect ones and design VSGD to prioritize more stable trajectories. Evaluations on GSM8K, MATH500, and an MMLU subset demonstrate consistent accuracy improvements and reduced incomplete reasoning, showing that modeling temporal reward dynamics enhances guided decoding efficiency and reliability.

**Strengths:**

The proposed method itself is coherent and easy to follow. The authors conduct experiments on several benchmarks to validate the effectiveness of the proposed method.

**Weaknesses:**

1.	Overall, the contribution of this paper, while meaningful, is somewhat incremental relative to prior PRM-guided decoding research. The method reuses standard components such as beam search and PRM scoring, with limited theoretical advancement beyond empirical validation.
2.	Across datasets, performance gains over the strongest baseline are relatively small (about 1–1.5% on average). Such improvements may not justify introducing an additional volatility computation layer in practical systems. Moreover, the improvements vary across domains, with some categories (e.g., Virology and GSM8K) showing minimal or even negative differences, indicating limited robustness.
3.	The authors employ only one LLM backbone in their experiments. To strengthen the evaluation, recently proposed LLMs, such as Mistral and Qwen should be included as the backbone models for comparation.
4.	The paper does not investigate how sensitive VSGD is to its hyperparameters, such as the stability constant ϵ or the aggregation function used in ranking candidates. It also lacks ablation experiments isolating the effects of volatility scaling from other algorithmic factors.

**Questions:**

1.	How sensitive is VSGD to the choice of the stability constant ϵ and the aggregation function Agg?
2.	Would volatility normalization still help when PRM rewards are poorly calibrated or highly correlated with token length?
3.	Could volatility be exploited during PRM training rather than only inference?
4.	Have the authors tested whether volatility correlates with interpretability or logical consistency of reasoning paths beyond correctness?

---

> ### Author Response · Authors · 2025-11-19
>
> We thank Reviewer ZvW3 for the positive remarks that the proposed method is coherent and easy to follow. We address the reviewer’s remaining concerns in detail below.
>
> > `[R4-1] Concern: Limited improvement magnitude across datasets`
>
> To address the reviewer's concern regarding the limited improvement magnitude, we ran additional experiments to assess whether our proposed VSGD can provide larger gains under challenging settings. We compared VSGD with guided decoding (GD) algorithm whlie keeping all hyperparameters indentical across methods. In particular, both GD-Last and VSGD-Last denote variants that use the Last aggregation function. Our evaluation uses the AMC23 dataset, reporting Pass@1 and Accuracy.
>
> `[Table T4-1: Performances of baseline method and our VSGD on AMC23]`
>
> | Dataset     | Method        |   Pass@1 |   Accuracy|
> |:------------|:--------------|---------:|----------:|
> | AMC23       | GD-Last       |     19.3 |      20.0 |
> | AMC23       | VSGD-Last     |     23.9 |      25.0 |
>
> As shown above, VSGD-Last achieves improvements of +4.6 Pass@1 and +5 Accuracy over GD-Last. This larger gain on AMC23 indicates that VSGD can offer more substantial advantages.
>
> We will include these new results in the revised manuscript.
>
>
> > `[R4-2] Concern: Computational overhead of volatility scaling`
>
> To address the concern regarding computational overhead, we measured the wall-clock runtime of our method, focusing in particular on the additional cost introduced by the volatility-based reward scaling step.
>
> When VSGD-Min is tested on MATH500 dataset, the table below reports two values:
> * (1) Total time — the overall running time of our method.
> * (2) Scaling step time — the time spent computing volatility and applying volatility-based reward scaling.
>
> `[Table T4-2: Total runtime and the time spent on the volatility-based scaling step]`
>
>
> |Total time(s) | Scaling step time (s)|   Scaling step time / Total time (%) |
> |--------:|------------------:|----------------:|
> | 165.51  |       0.000821817 |      0.0005 |
>
> As shown in the above table, the scaling step accounts for only 0.0005% of the total runtime. These results indicate that volatility scaling introduces negligible computational overhead with no practical impact on efficiency.
>
> We will include these new results in the revised manuscript.
>
> > `[R4-3] Generalizability to recently proposed LLM backbone`
>
> To address the reviewer’s concern, we conducted additional experiments demonstrating that the key findings reported for the `LLaMa` model in our submitted manuscript—namely the volatility behavior and performance gains—consistently extend to the latest high-performing LLMs under the same experimental setup.
>
> **Volatility behavior in `GPT-4o-mini`**
>
> We first replaced the original backbone LLM with GPT-4o-mini while keeping all other components and hyperparameters fixed. We then evaluated volatility patterns on the AMC23 dataset. As illustrated in the figure below, the reward trajectories of correct reasoning paths exhibit lower volatility than those of incorrect reasoning paths. This behavior is consistent with what we have reported for `LLaMa` model in our submitted manuscript.
>
> [Figure F4-1: Volatility distribution when GPT-4o-min is used](https://hackmd.io/_uploads/HJcxwfugWl.png)
>
> **Performance comparison for `GPT-4o-mini`: VSGD-Min vs GD-Min**
>
> Next, we evaluate VSGD-Min against GD-Min, where GD-Min shares all hyperparameters with VSGD-Min but does not apply volatility scaling. We test both methods on AMC23 and AIME2024, reporting Pass@1.
>
> `[Table T4-3: Performances of baseline method and our VSGD when GPT-4o-mini is used]`
>
> | Dataset   | Method        |   Pass@1 |
> |:----------|:--------------|---------:|
> | AMC23     | GD-Min        |     13.1 |
> | AMC23     | VSGD-Min      |     13.4 |
> | AIME2024  | GD-Min        |      8.8 |
> | AIME2024  | VSGD-Min      |      9.2 |
>
> As shown in the table above, VSGD-Min consistently outperforms GD-Min on both datasets (+0.3 on AMC23 and +0.4 on AIME2024). While modest, these gains are consistent across datasets and provide additional evidence that volatility scaling offers measurable gains when applied to GPT-based models.
>
> We will include these new results in the revised manuscript.

---

> ### Author Response · Authors · 2025-11-19
>
> > `[R4-4] Concern : Hyperparameter sensitivity`
>
> We thank the reviewer for raising important questions.
>
> In the submitted manuscript, Algorithm 2 outlines the main hyperparameters of VSGD, including
> * the maximum number of reasoning steps $T$
> * the number of sampled reasoning paths $L$.
> * aggregation function (`Min`, `Sum`, `Last`)
> * stability constant $\epsilon$
>
>
> Among these four hyperparameters, the effect of the first two hyperparameters ($T$ and $L$) are already analyzed in Figures 3 and 4 in the submitted manuscript. Figure 3 shows that our method consistently outperforms the baselines regardless of $T$, while Figure 4 demonstrates that it also surpasses the baselines when $L$ is sufficiently large.
>
>
> As per the reviewer's comment, we ran additional experiments showing the sensitivity of our method on other two hyperparameters: the aggregation function and the stability constant $\epsilon$.
>
> **Sensitivity to the `Aggregation Function`**
>
> The below table reports the performances of GD (baseline) and VSGD (ours) for three aggregation functions (`Min`, `Sum`, `Last`) on MMLU-Pro dataset, using LLaMA as the LLM and VersaPRM as the PRM.
>
> `[Table T4-4: Performances of baseline method and ours under three different aggregation functions]`
>
>
> |   GD-Sum |   GD-Last |   GD-Min |   VSGD-Sum |   VSGD-Last |   VSGD-Min |
> |---------:|----------:|---------:|-----------:|------------:|-----------:|
> |    34.76 |     36.14 |    36.31 |      36.51 |       36.58 |      36.78 |
>
> As shown in the table above, VSGD improves upon each GD variant by +1.75 (`Sum`), +0.44 (`Last`), and +0.47 (`Min`), with VSGD-Min achieving the highest overall accuracy. These results indicate that volatility scaling provides consistent benefits regardless of the aggregation scheme.
>
>
> **Sensitivity to the `Stability Constant` $\epsilon$**
>
> The below table compares the performances of VSGD (ours) on MATH500 dataset for three stability constants $\epsilon\in\{10^{-4}, 10^{-5}, 10^{-6}\}$, using LLaMA as the LLM and VersaPRM as the PRM.
>
> `[Table T4-5: Performances of our VSGD under three different stability constant]`
>
> | Method                |   Accuracy |
> |:----------------------|----------:|
> | VSGD($\epsilon=10^{-4}$) |      51.0 |
> | VSGD($\epsilon=10^{-5}$) |      51.0 |
> | VSGD($\epsilon=10^{-6}$) |      51.7 |
>
> As shown in the table above, VSGD can be considered to be insensitive to the choice of $\epsilon$; the performance remains stable across values ranging from $10^{-4}$ to $10^{-6}$.
>
> We will include these new results in the revised manuscript.

---

> ### Author Response · Authors · 2025-11-19
>
> > `[R4-5] Questions on the behavior when PRM scores are less calibrated or correlated with token length`
>
> As per the reviewer's question, we conducted additional experiments showing that the proposed VSGD method still help when PRM rewards are less calibrated or correlated with token length, details of which are shown below.
>
> `#1. Experiments when PRM rewards are less calibrated`
>
> We conducted additional experiments using a PRM that is intentionally mismatched with the target domain. To examine this setup, we use QwenPRM800K [1], which is trained exclusively on mathematical reasoning data. We replace the PRM used in the submitted manuscript with QwenPRM800K and apply it to a MMLU-Pro dataset. In particular, we follow the categorization in [2] which partitions domains into three groups:
> * (1) Math
> * (2) Math-Adjacent (Chemistry, Computer science, Engineering and Physics)
> * (3) Non-Math (Biology, Business, Economics, Health, History, Law, Philosophy, Psychology and Other)
>
> The below table reports the performances of GD-Min (baseline) and VSGD-Min (ours) for three groups.
>
> `[Table T4-6: Performances of our VSGD when PRM scores are less calibrated]`
>
> | Group         |   GD-Min |   VSGD-Min |
> |:--------------|---------:|-----------:|
> | Math          |    18.36 |      19.10 |
> | Math Adjacent |    20.02 |      21.28 |
> | Non-Math      |    40.73 |      39.06 |
>
>
> VSGD-Min shows consistent improvements over the GD baselines in the Math and Math-Adjacent groups, indicating that volatility-based guidance remains beneficial when the PRM retains partial task relevance despite domain shift.
> In contrast, performance in the Non-Math group decreases, which is expected because the math-oriented PRM exhibits weaker calibration in these domains and introduces noises in volatility computation.
> These results suggest that VSGD exhibits partial robustness under degraded PRM calibration but its effectiveness depends on the degree of alignment between the PRM and the target domain. We will make this dependency explicit in the revised version of the manuscript.
>
> `#2. Experiments when PRM rewards are correlated with token length`
>
> We additionally analyze whether token length affects PRM scores and whether such effects interact with the performance of VSGD. Using MATH500 dataset, we measure the correlation between the token length of the generated reasoning path and its aggregated PRM score for two different PRMs (VersaPRM and RLHFlow-PRM-DeepSeek-8B [3]), alongside the corresponding accuracies under GD-Min (baseline) and VSGD-Min (ours).
>
> `[Table T4-7: Performances of our VSGD when PRM scores are correlated with token length]`
>
> | PRM                      | Method   | Corr.    | Acc. |
> | ------------------------ | -------- | ------- | --- |
> | VersaPRM                 | GD-Min   | -0.5014 | 50.1|
> | VersaPRM                 | VSGD-Min | -0.5090 | 50.8|
> | RLHFlow-PRM-DeepSeek-8B  | GD-Min   | -0.4413 | 46.3|
> | RLHFlow-PRM-DeepSeek-8B  | VSGD-Min | -0.4185 | 47  |
>
> The above table shows that VSGD-Min yields higher accuracy in both the stronger-correlation (VersaPRM) and weaker-correlation (RLHFlow) settings. These results indicate that VSGD remains beneficial even when PRM scores are influenced by length-related biases.
>
> We will include these new results in the revised manuscript.
>
>
> [1] https://huggingface.co/UW-Madison-Lee-Lab/Qwen-PRM800K
>
> [2] Zeng, Thomas, et al. "Versaprm: Multi-domain process reward model via synthetic reasoning data." arXiv preprint arXiv:2502.06737 (2025).
>
> [3] https://huggingface.co/RLHFlow/Llama3.1-8B-PRM-Deepseek-Data

---

> ### Author Response · Authors · 2025-11-19
>
> > `[R4-6] Questions on correlation of volatility to other reasoning properties`
>
> To directly address the reviewer’s question, we find that volatility is moderately correlated with another key reasoning property—logical consistency.
>
> We first annotated the logical consistency of each generated reasoning path with respect to the ground-truth solution. For annotation, we adopted the established prompt framework from Ghosh et al. [1], which assesses whether reasoning path is logically consistent with the correct solution.
>
> Using the same experimental setup as in Figure 2 of the submitted manuscript, we measure the relationship between volatility and logical consistency on GSM8K and MATH500 datasets, the results of which are given below.
>
> [Figure F4-2: Volatility distribution across consistency groups on GSM8K](https://hackmd.io/_uploads/H1G6ViYeWl.png)
>
> [Figure F4-3: Volatility distribution across consistency groups on MATH500](https://hackmd.io/_uploads/rJWT4sYlWe.png)
>
> Across both datasets, the inconsistent group exhibits a clear rightward shift toward higher volatility.
>
> We will include these new results in the revised manuscript.
>
> [1] Ghosh, Bishwamittra, et al. "Logical consistency of large language models in fact-checking." arXiv preprint arXiv:2412.16100 (2024).
>
> > `[R4-7] Question on potential use of volatility in PRM training`
>
> We thank the reviewer for raising this interesting direction. We agree that volatility could provide an additional supervision signal during PRM training. In particular, volatility may help identify reasoning paths that exhibit unstable or internally inconsistent reward patterns, complementing existing approaches that filter or weight paths based on logical consistency [1]. Such auxiliary signals—whether derived from consistency checks or volatility—could potentially be used to prioritize more reliable trajectories or down-weight noisy ones. While we view this as a promising extension, a thorough integration of volatility into PRM training is beyond the scope of the current work and we plan to explore it in future research.
>
> [1] Feng, Yu, et al. "VeriCoT: Neuro-symbolic Chain-of-Thought Validation via Logical Consistency Checks." arXiv preprint arXiv:2511.04662 (2025).
>
>
> > `[R4-8] Concerns on theoretical depth`
>
> We appreciate the reviewer’s comments regarding the theoretical grounding of our approach. We agree that our manuscript does not provide a mathematical statement proving the effectiveness of our method. However, our statistical analysis shows that correct and incorrect reasoning trajectories exhibit systematically different volatility patterns; correct trajectories tend to produce more stable reward dynamics, whereas incorrect ones display higher dispersion. This suggests that *volatility* provides an additional reliability signal that can complement decoding algorithms relying solely on reward values. We view this empirical observation as an important first step towards understanding and improving the LLM decoding process.

---

### Official Review · Reviewer_m9QW · 2025-10-29

**Soundness:** 2
**Presentation:** 2
**Contribution:** 2
**Rating:** 2
**Confidence:** 4

**Summary:**

This paper addresses the instability of reward signals in Process Reward Model (PRM)–guided decoding for large language models. The authors observe that correct reasoning paths exhibit stable (low-volatility) reward trajectories, while incorrect ones fluctuate significantly. To exploit this, they propose Volatility-Scaled Guided Decoding (VSGD), which adjusts reward values based on their temporal volatility to favor stable reasoning paths. Experiments on GSM8K, MATH500, and MMLU subsets show that VSGD improves reasoning accuracy and reduces incomplete reasoning, demonstrating that incorporating reward stability enhances guided decoding performance.

**Strengths:**

The paper’s strength lies in introducing Volatility-Scaled Guided Decoding (VSGD), which leverages reward stability to guide search. By prioritizing reasoning paths with low reward volatility, it offers a simple yet effective way to stabilize decoding and improve reasoning accuracy.

**Weaknesses:**

1. The paper only observes the fluctuation of PRM rewards empirically without providing a theoretical explanation.
It remains unclear why reward volatility naturally emerges or correlates with reasoning correctness.

2. All experiments rely solely on LLaMA-3.1-8B and VersaPRM, making the results model-specific.
It is uncertain whether stronger models like GPT-5 would show the same volatility behavior or performance gains.

3. The study omits stronger decoding and test-time scaling baselines such as MCTS or Q-function-based methods. Moreover, the reported accuracy improvement is small (around +1.1–1.4), limiting the practical significance.

4. The paper introduces several hyperparameters but does not analyze their sensitivity or robustness. Without such evaluation, it is difficult to assess the stability and reproducibility of the proposed method.

**Questions:**

see weaknesses

---

> ### Author Response · Authors · 2025-11-19
>
> We thank the reviewer m9QW for the thoughtful assessment and appreciate the recognition that VSGD provides a simple and effective way to stabilize decoding by leveraging reward stability. We are encouraged that the motivation and empirical observations underlying our approach were viewed as clear and meaningful. We value these positive remarks and address the reviewer’s concerns in detail below.
>
> > `[R3-1] Generalizability of our results to stronger LLM`
>
> To address the reviewer’s concern, we conducted additional experiments demonstrating that the key findings reported for the `LLaMa` model in our submitted manuscript—namely the volatility behavior and performance gains—consistently extend to the latest high-performing LLMs under the same experimental setup.
>
> **Volatility behavior in `GPT-4o-mini`**
>
> We first replaced the original backbone LLM with GPT-4o-mini while keeping all other components and hyperparameters fixed. We then evaluated volatility patterns on the AMC23 dataset. As illustrated in the figure below, the reward trajectories of correct reasoning paths exhibit lower volatility than those of incorrect reasoning paths.
>
> [Figure F3-1: Volatility distribution when GPT-4o-min is used](https://hackmd.io/_uploads/HJcxwfugWl.png)
>
> **Performance comparison for `GPT-4o-mini`: VSGD-Min vs GD-Min**
>
> Next, we evaluate VSGD-Min against GD-Min, where GD-Min shares all hyperparameters with VSGD-Min but does not apply volatility scaling. We test both methods on AMC23 and AIME2024, reporting Pass@1.
>
> `[Table T3-1: Performances of baseline method and our VSGD when GPT-4o-mini is used]`
>
> | Dataset   | Method        |   Pass@1 |
> |:----------|:--------------|---------:|
> | AMC23     | GD-Min        |     13.1 |
> | AMC23     | VSGD-Min      |     13.4 |
> | AIME2024  | GD-Min        |      8.8 |
> | AIME2024  | VSGD-Min      |      9.2 |
>
> As shown in the table above, VSGD-Min consistently outperforms GD-Min on both datasets (+0.3 on AMC23 and +0.4 on AIME2024). While modest, these gains are consistent across datasets and provide additional evidence that volatility scaling offers measurable gains when applied to GPT-based models.
>
> We will include these new results in the revised manuscript.

---

> ### Author Response · Authors · 2025-11-19
>
> > `[R3-2] Generalizability to stronger decoding algorithms`
>
> As suggested by the reviewer, we ran additional experiments to demonstrate that the proposed Volatility-Scaled Guided Decoding (VSGD) method can be extended to other tree-based decoding algorithms such as MCTS. We first describe how volatility scaling is incorporated into MCTS and then present the experimental results.
>
> In MCTS, each partial solution corresponds to a node in the search tree, and the PRM provides step-level rewards that guide the node selection. We consider two methods :
> - **MCTS-GD (Baseline)** : the original MCTS with PRM guidance, where raw PRM rewards are directly used during the node selection.
> - **MCTS-VSGD (Ours)** : identical to MCTS-GD except that raw PRM rewards are replaced with volatility-scaled rewards in the node-selection rule.
>
> We compare MCTS-GD with MCTS-VSGD using Llama-8B as the LLM and VersaPRM as the PRM. For both MCTS-GD and MCTS-VSGD, we evaluate three different aggregation functions used to compute node values during search, namely, `Sum`, `Last` and `Min`. Experiments are conducted on the MATH500 benchmark, and we report Accuracy and Pass@1.
>
> `[Table T3-2: Performances of baseline method and ours using Sum aggregation function]`
>
> | Method         |   Pass@1 |   Accuracy|
> |:---------------|---------:|----------:|
> | MCTS-GD-Sum    |     43.1 |      46.2 |
> | MCTS-VSGD-Sum  |     44.6 (`+1.5`) |      48.2(`+2.0`) |
>
> `[Table T3-3: Performances of baseline method and ours using Min aggregation function]`
>
> | Method         |   Pass@1 |   Accuracy|
> |:---------------|---------:|----------:|
> | MCTS-GD-Min    |     44.6 |      52.4 |
> | MCTS-VSGD-Min  |     45.6 (`+1.0`) |      52.4(+0.0) |
>
> `[Table T3-4: Performances of baseline method and ours using Last aggregation function]`
>
> | Method         |   Pass@1 |   Accuracy|
> |:---------------|---------:|----------:|
> | MCTS-GD-Last   |     45.6 |      52.8 |
> | MCTS-VSGD-Last |     45.9(`+0.3`) |      54.0(`+1.2`) |
>
> Across all aggregation functions, MCTS-VSGD provides consistent improvements over MCTS-GD, indicating that volatility scaling generalizes effectively to MCTS-based decoding.
>
> We will include these new results in the revised manuscript.
>
> > `[R3-3] Concern: Limited improvement magnitude across datasets`
>
> To address the reviewer's concern regarding the limited improvement magnitude, we ran additional experiments to assess whether our proposed VSGD can provide larger gains under challenging settings. We compared VSGD with guided decoding (GD) algorithm whlie keeping all hyperparameters indentical across methods. In particular, both GD-Last and VSGD-Last denote variants that use the Last aggregation function. Our evaluation uses the AMC23 dataset, reporting Pass@1 and Accuracy.
>
> `[Table T3-5: Performances of baseline method and ours on AMC23 dataset]`
>
>
> | Dataset     | Method        |   Pass@1 |   Accuracy|
> |:------------|:--------------|---------:|----------:|
> | AMC23       | GD-Last       |     19.3 |      20.0 |
> | AMC23       | VSGD-Last     |     23.9 |      25.0 |
>
> As shown above, VSGD-Last achieves improvements of +4.6 Pass@1 and +5 Accuracy over GD-Last. This larger gain on AMC23 indicates that VSGD can offer more substantial advantages.
>
> We will include these new results in the revised manuscript.

---

> ### Author Response · Authors · 2025-11-19
>
> > `[R3-4] Concerns on hyperparameter analysis`
>
> We thank the reviewer for raising important questions.
>
> In the submitted manuscript, Algorithm 2 outlines the main hyperparameters of VSGD, including
> * the maximum number of reasoning steps $T$
> * the number of sampled reasoning paths $L$.
> * aggregation function (`Min`, `Sum`, `Last`)
> * stability constant $\epsilon$
>
>
> Among these four hyperparameters, the effect of the first two hyperparameters ($T$ and $L$) are already analyzed in Figures 3 and 4 in the submitted manuscript. Figure 3 shows that our method consistently outperforms the baselines regardless of $T$, while Figure 4 demonstrates that it also surpasses the baselines when $L$ is sufficiently large.
>
>
> As per the reviewer's comment, we ran additional experiments showing the sensitivity of our method on other two hyperparameters: the aggregation function and the stability constant $\epsilon$.
>
> **Sensitivity to the `Aggregation Function`**
>
> The below table reports the performances of GD (baseline) and VSGD (ours) for three aggregation functions (`Min`, `Sum`, `Last`) on MMLU-Pro dataset, using LLaMA as the LLM and VersaPRM as the PRM.
>
> `[Table T3-6: Performances of baseline method and ours under three different aggregation functions]`
>
> |   GD-Sum |   GD-Last |   GD-Min |   VSGD-Sum |   VSGD-Last |   VSGD-Min |
> |---------:|----------:|---------:|-----------:|------------:|-----------:|
> |    34.76 |     36.14 |    36.31 |      36.51 |       36.58 |      36.78 |
>
> As shown in the table above, VSGD improves upon each GD variant by +1.75 (`Sum`), +0.44 (`Last`), and +0.47 (`Min`), with VSGD-Min achieving the highest overall accuracy. These results indicate that volatility scaling provides consistent benefits regardless of the aggregation scheme.
>
>
> **Sensitivity to the `Stability Constant` $\epsilon$**
>
> The below table compares the performances of VSGD (ours) on MATH500 dataset for three stability constants $\epsilon\in\{10^{-4}, 10^{-5}, 10^{-6}\}$, using LLaMA as the LLM and VersaPRM as the PRM.
>
> `[Table T3-7: Performances of our VSGD under three different stability constant]`
>
> | Method                |   Accuracy |
> |:----------------------|----------:|
> | VSGD($\epsilon=10^{-4}$) |      51.0 |
> | VSGD($\epsilon=10^{-5}$) |      51.0 |
> | VSGD($\epsilon=10^{-6}$) |      51.7 |
>
> As shown in the table above, VSGD can be considered to be insensitive to the choice of $\epsilon$; the performance remains stable across values ranging from $10^{-4}$ to $10^{-6}$.
>
> We will include these new results in the revised manuscript.
>
>
> > `[R3-5] Concerns on theoretical depth`
>
> We appreciate the reviewer’s comments regarding the theoretical grounding of our approach. We agree that our manuscript does not provide a mathematical statement proving the effectiveness of our method. However, our statistical analysis shows that correct and incorrect reasoning trajectories exhibit systematically different volatility patterns; correct trajectories tend to produce more stable reward dynamics, whereas incorrect ones display higher dispersion. This suggests that *volatility* provides an additional reliability signal that can complement decoding algorithms relying solely on reward values. We view this empirical observation as an important first step towards understanding and improving the LLM decoding process.

---

### Official Review · Reviewer_EJpL · 2025-11-01

**Soundness:** 3
**Presentation:** 3
**Contribution:** 2
**Rating:** 4
**Confidence:** 3

**Summary:**

The authors propose a new decoding algorithm for Large Language Models (LLMs) called Volatility-Scaled Guided Decoding (VSGD), which prioritizes candidate paths with lower volatility by jointly considering the magnitude of PRM-estimated rewards and the volatility of these rewards across decoding step. The technique is derived from the observation that correct reasoning paths usually exhibit high rewards but low volatility across reasoning steps. Thus, instead of selecting candidates with maximum reward in each step, the authors select the ones with the maximum ratio of reward and standard deviation. Experimental results demonstrate the merits of the proposed method.

**Strengths:**

- In terms of clarity, I think the authors did a great job. From pilot study to algorithm, everything is crystal clear.
- I think exploiting the dynamics of reward for better decoding is interesting and promising. The use of std as a damping factor avoids myopic emphasis on high-reward steps.
- The experiments are comprehensive.

**Weaknesses:**

- The technical depth is limited. The major contribution is two folds: (i) the empirical observation of PRM dynamics of both correct/incorrect reasoning paths; (ii) replacing the selection criterion in beam search to incoprate variability with no theoretical result, which seem to be straightfoward. Also, it is unclear to me that whether the proposed method can be generalized to other decoding algorithms like MCTS.
- The experimental results are not strong and convincing enough. The authors only compared their methods with vanilla beam search which is obviously not the SotA methods for test-time scaling. The datasets used are also a bit old. I suggest the authors considering popular benchmarks like AIME or AMC. Finally, the improvement is marginal, which sheds some doubts on the significance of the proposed method.

**Questions:**

See Weakness section

---

> ### Author Response · Authors · 2025-11-19
>
> We thank the reviewer EJpL for the thoughtful evaluation and for recognizing the clarity of our motivation, pilot study, and algorithmic formulation. We are also encouraged that the reviewer considers leveraging reward dynamics for decoding to be an interesting and promising direction, and we appreciate the acknowledgment that our experiments are comprehensive. We appreciate your careful reading of the submission and address your comments below.
>
> > `[R2-1] Generalizability of proposed method to other decoding algorithms`
>
> As suggested by the reviewer, we ran additional experiments to demonstrate that the proposed Volatility-Scaled Guided Decoding (VSGD) method can be extended to other tree-based decoding algorithms such as MCTS. We first describe how volatility scaling is incorporated into MCTS and then present the experimental results.
>
> In MCTS, each partial solution corresponds to a node in the search tree, and the PRM provides step-level rewards that guide the node selection. We consider two methods :
> - **MCTS-GD (Baseline)** : the original MCTS with PRM guidance, where raw PRM rewards are directly used during the node selection.
> - **MCTS-VSGD (Ours)** : identical to MCTS-GD except that raw PRM rewards are replaced with volatility-scaled rewards in the node-selection rule.
>
> In the tables (T2-1, T2-2 and T2-3) below, we compare MCTS-GD with MCTS-VSGD using Llama-8B as the LLM and VersaPRM as the PRM. For both MCTS-GD and MCTS-VSGD, we evaluate three different aggregation functions used to compute node values during search, namely, `Sum`, `Last` and `Min`. Experiments are conducted on the MATH500 benchmark, and we report accuracy and Pass@1.
>
> `[Table T2-1: Performances of baseline method and ours using Sum aggregation function]`
>
> | Method         |   Pass@1 |   Accuracy|
> |:---------------|---------:|----------:|
> | MCTS-GD-Sum    |     43.1 |      46.2 |
> | MCTS-VSGD-Sum  |     44.6 (`+1.5`) |      48.2(`+2.0`) |
>
> `[Table T2-2: Performances of baseline method and ours using Min aggregation function]`
>
> | Method         |   Pass@1 |   Accuracy|
> |:---------------|---------:|----------:|
> | MCTS-GD-Min    |     44.6 |      52.4 |
> | MCTS-VSGD-Min  |     45.6 (`+1.0`) |      52.4(+0.0) |
>
> `[Table T2-3: Performances of baseline method and ours using Last aggregation function]`
>
> | Method         |   Pass@1 |   Accuracy|
> |:---------------|---------:|----------:|
> | MCTS-GD-Last   |     45.6 |      52.8 |
> | MCTS-VSGD-Last |     45.9(`+0.3`) |      54.0(`+1.2`) |
>
> Across all aggregation functions, MCTS-VSGD provides consistent improvements over MCTS-GD, indicating that volatility scaling generalizes effectively to MCTS-based decoding.
>
> We will include these new results in the revised manuscript.

---

> ### Author Response · Authors · 2025-11-19
>
> > `[R2-2] Comparison with SOTA test time scaling method`
>
> To address the reviewer's suggestion, we ran additional experiments using commonly adopted test-time scaling baselines such as Best-of-N (with $N=10$) and Self-Consistency [1], as well as guided decoding algorithms based on beam search (BS-GD) [2] and MCTS (MCTS-GD) [3]. We then compare these methods with our volatility-scaled guided decoding algorithms, BS-VSGD and MCTS-VSGD, which incorporate volatility-scaled PRM rewards into beam search and MCTS, respectively.
>
> The experimental results are shown in `Table T2-4` below. Here, we conduct experiments on the MATH500 benchmark, using VersaPRM as the process reward model and Llama-8B as the backbone LLM. For both beam search and MCTS, we consider three reward aggregation functions—`Last` and `Min`.
>
>
> `[Table T2-4: Performances of test time scaling methods on MATH500 benchmark]`
>
> | Method           |   Pass@1 |   Accuracy|
> |:-----------------|---------:|----------:|
> | Best-of-N        |     39.0 |     42.6 |
> | Self Consistency |     39.0 |     50.6 |
> | BS-GD-Last       |     48.6 |     50.4 |
> | BS-VSGD-Last     |     46.9 |     49.6 |
> | BS-GD-Min        |     48.3 |     50.8 |
> | BS-VSGD-Min      |    **49.3** |     **51.0** |
> | MCTS-GD-Min      |     44.6 |     52.4 |
> | MCTS-VSGD-Min    |     45.6 |     52.4 |
> | MCTS-GD-Last     |     45.6 |     52.8 |
> | MCTS-VSGD-Last   |     **45.9** |     **54.0** |
>
> Across both beam-search and MCTS-based decoding, the volatility-scaled variants generally provide improvements over the corresponding guided-decoding baselines.
>
> For example, BS-VSGD-Min achieves the highest accuracy (51.0) among beam search-based methods, while MCTS-VSGD-Last yields the highest overall accuracy (54.0) among MCTS-based methods. These results suggest that incorporating volatility-scaled rewards can provide a complementary signal during decoding, enhancing reasoning performance compared to widely used test-time scaling methods.
>
> We will include these new results in the revised manuscript.
>
>
> [1] Wang, Xuezhi, et al. "Self-consistency improves chain of thought reasoning in language models." arXiv preprint arXiv:2203.11171 (2022).
>
> [2] Snell, Charlie, et al. "Scaling llm test-time compute optimally can be more effective than scaling model parameters." arXiv preprint arXiv:2408.03314 (2024).
>
> [3] Feng, Xidong, et al. "Alphazero-like tree-search can guide large language model decoding and training." arXiv preprint arXiv:2309.17179 (2023).
>
> > `[R2-3] Evalution on additional dataset`
>
> Following the reviewer’s suggestion, we conducted additional evaluations on two challenging mathematical reasoning benchmarks: AMC23 and MinervaMath.
> We compare our proposed VSGD with the beam search–based guided decoding (GD) baseline under the exact same experimental setup used in our original submission. The results are shown below.
>
> `[Table T2-5: Performances of baseline method and our VSGD on AMC23 and MinervaMath]`
>
> | Dataset     | Method        |   Pass@1 |   Accuracy|
> |:------------|:--------------|---------:|----------:|
> | AMC23       | GD-Last       |     19.3 |      20.0 |
> | AMC23       | VSGD-Last     |     23.9 |      25.0 |
> | MinervaMath | GD-Last       |     11.3 |      12.1 |
> | MinervaMath | VSGD-Last     |     12.5 |      13.2 |
>
> Across both benchmarks, VSGD consistently achieves higher Pass@1 and accuracy than the GD baseline under matched settings. For example, on AMC23, VSGD yields improvements of +4.6 Pass@1 and +5.0 accuracy. These results suggest that VSGD can provide performance improvements on popular benchmarks.
>
> We will include these new results in the revised manuscript.

---

> ### Author Response · Authors · 2025-11-19
>
> > `[R2-4] Concerns on the marginal improvement`
>
>
> To address the reviewer's concern regarding the limited improvement magnitude, we ran additional experiments to assess whether our proposed VSGD can provide larger gains under challenging settings. We compared VSGD with guided decoding (GD) algorithm whlie keeping all hyperparameters indentical across methods. In particular, both GD-Last and VSGD-Last denote variants that use the `Last` aggregation function. Our evaluation uses the AMC23 dataset, reporting Pass@1 and Accuracy.
>
>
>
> `[Table T2-6: Performances of baseline method and our VSGD on AMC23]`
>
> | Dataset     | Method        |   Pass@1 |   Accuracy|
> |:------------|:--------------|---------:|----------:|
> | AMC23       | GD-Last       |     19.3 |      20.0 |
> | AMC23       | VSGD-Last     |     23.9 |      25.0 |
>
> As shown above, VSGD-Last achieves improvements of +4.6 Pass@1 and +5 Accuracy over GD-Last. This larger gain on AMC23 indicates that VSGD can offer more substantial advantages.
>
> We will include these new results in the revised manuscript.
>
> > `[R2-5] Concerns on theoretical depth`
>
> We appreciate the reviewer’s comments regarding the theoretical grounding of our approach. We agree that our manuscript does not provide a mathematical statement proving the effectiveness of our method. However, our statistical analysis shows that correct and incorrect reasoning trajectories exhibit systematically different volatility patterns; correct trajectories tend to produce more stable reward dynamics, whereas incorrect ones display higher dispersion. This suggests that *volatility* provides an additional reliability signal that can complement decoding algorithms relying solely on reward values. We view this empirical observation as an important first step towards understanding and improving the LLM decoding process.

---

> > ### Comment · Reviewer_EJpL · 2025-11-23
> >
> > Thanks for the authors' efforts for addressing my concerns. However, my skepticisms remain. In term of performance improvement, the majority of experiments (including the results in authors' response) show very minor improvements. In terms of technical contributions, I still think scaling PRM values with a running std is a useful trick for the community, but is not sufficient for top venue like ICLR, given no theoretical guarantees or significant performance boost. Therefore, I trend to keep my initial assessment.

---

### Official Review · Reviewer_kJFH · 2025-11-01

**Soundness:** 2
**Presentation:** 2
**Contribution:** 2
**Rating:** 2
**Confidence:** 3

**Summary:**

This paper focues on guided decoding with a process reward model. Firstly, the authors observe that the reward distribution for different steps has a close correlation with correctness. I.e. a declining reward across steps and a high volatility patten of the rewards normally hint a final false prediction. Inspired by this observation, the authors propose Volatility-Scaled Guided Decoding (VSGD), which prioritizes candidate paths with lower volatility by jointly considering the magnitude of PRM-estimated rewards and the volatility of these rewards across decoding steps.

Experiments on three benchmarks (GSM8K, MATH500 and MMLU) with a process reward model show that VSGD outperforms other baselines. Extensive ablation also justify the design choice.

**Strengths:**

1. The observation is interesting. I.e. reward stability and volatility distribution is correlated with the prediction correctness.
2. The proposed method is closely related to the observation, making the paper coherent.
3. Extensive experimemts and ablation show the benefits from VSGD.

**Weaknesses:**

1. Lack of process reward models and decoding models. Only one process reward model (VersaPRM) and one decoding model (Llama-8B) are verified here. It cast doubts on the generalization of the observation and results.
2. Limited improvement. From Table 1, we can see that the improvement from VSGD is very limited compared, +0.6 or +0.7 on MATH500 and -0.2 or +0.1 on GSM8K. The most improvement comes form MMLU, with +1.1 or +1.4. This shows the limitation of the proposed method on various domains.

**Questions:**

None

---

> ### Author Response · Authors · 2025-11-19
>
> We thank Reviewer kJFH for recognizing that our empirical observation on reward stability and volatility is interesting, that our proposed VSGD is well aligned with this observation, and that our experiments and ablations help support the design choices of VSGD. We appreciate your careful reading of the submission and address your comments below.
>
> > `[R1-1] Concerns regarding the limited diversity of PRMs and LLMs`
>
> As per the reviewer's concern, we conducted additional experiments showing that the proposed VSGD method outperforms baselines across various PRMs and LLMs, details of which are shown below.
>
> First, to address the concern regarding PRM diversity, we evaluated VSGD under an alternative reward model by replacing the VersaPRM used in the submitted version with `RLHFlow-PRM-DeepSeek-8B (RLHFlow-PRM)` [1]. Since RLHFlow-PRM is tailored for mathematical reasoning, we conducted experiments on MATH500 and GSM8K. The below table compares the baseline guided decoding (GD) method and our VSGD method, when RLHFlow-PRM is used instead of VersaPRM.
>
>
> `[Table T1-1: Performances of baseline method (GD) and ours when RLHFlow-PRM is used]`
>
> | Method         |   MATH500 |   GSM8K |   Average |
> |:---------------|----------:|--------:|----------:|
> | GD (baseline)  |      46.3 |    86.7 |      66.5 |
> | VSGD   (ours)  |  **47**   |**87**  |  **67**   |
>
> As shown above, VSGD consistently outperforms GD (+0.7 on MATH500 and +0.3 on GSM8K). These results indicate that our VSGD remains effective even when paired with a different PRM (RLHFlow-PRM).
>
> Second, to address the concern regarding LLM diversity, we evaluated VSGD using `GPT-4o-mini` as an alternative LLM. Using VersaPRM as the PRM, we assessed performance on the AMC23 and AIME2024, reporting Pass@1.
>
> `[Table T1-2: Performances of baseline method (GD) and our VSGD when GPT-4o-mini is used]`
>
> | Dataset   | Method        |   Pass@1 |
> |:----------|:--------------|---------:|
> | AMC23     | GD-Min        |     13.1 |
> | AMC23     | VSGD-Min      |     13.4 |
> | AIME2024  | GD-Min        |      8.8 |
> | AIME2024  | VSGD-Min      |      9.2 |
>
> As shown in the table above, VSGD-Min consistently outperforms GD-Min on both datasets (+0.3 on AMC23 and +0.4 on AIME2024). While modest, these gains are consistent across datasets and provide additional evidence that volatility scaling offers measurable gains when applied to GPT-4o-mini.
>
> We will include these new results in the revised manuscript.
>
>
> [1] https://huggingface.co/RLHFlow/Llama3.1-8B-PRM-Deepseek-Data
>
> [2] Y. Wang, X. Ma, G. Zhang, Y. Ni, A. Chandra, S. Guo, and W. Chen. MMLU-Pro: A More Robust and Challenging Multi-Task Language Understanding Benchmark. Advances in Neural Information Processing Systems (NeurIPS), 37:95266–95290, 2024.
>
>
> ---
>
>
> > `[R1-2] Concern: Limited improvement magnitude across datasets`
>
> To address the reviewer's concern regarding the limited improvement magnitude, we ran additional experiments to assess whether our proposed VSGD can provide larger gains under challenging settings. We compared VSGD with guided decoding (GD) algorithm whlie keeping all hyperparameters indentical across methods. In particular, both GD-Last and VSGD-Last denote variants that use the Last aggregation function. Our evaluation uses the AMC23 dataset, reporting Pass@1 and Accuracy.
>
> `[Table T1-3: Performances of baseline method and ours on AMC23 dataset]`
>
>
> | Dataset     | Method        |   Pass@1 |   Accuracy|
> |:------------|:--------------|---------:|----------:|
> | AMC23       | GD-Last       |     19.3 |      20.0 |
> | AMC23       | VSGD-Last     |     23.9 |      25.0 |
>
> As shown above, VSGD-Last achieves improvements of +4.6 Pass@1 and +5 Accuracy over GD-Last. This larger gain on AMC23 indicates that VSGD can offer more substantial advantages.
>
> We will include these new results in the revised manuscript.

---

### Meta-Review · Area_Chair_64Gj · 2026-01-02

**Summary:**

The reviewers generally agree that the paper is clearly written and studies a sensible empirical phenomenon in PRM-guided decoding, namely that correct reasoning paths tend to exhibit more stable reward trajectories than incorrect ones. The proposed method, volatility-scaled guided decoding VSGD, is simple and well motivated by this observation, and the experimental results consistently show small improvements over standard PRM-guided decoding baselines. At the same time, multiple reviewers raised concerns that the contribution is incremental and largely heuristic in nature, with no theoretical justification and relatively modest performance gains on the main benchmarks. While the rebuttal added many additional experiments (more models, PRMs, decoding variants, and datasets), these results mostly reinforce the same pattern of small improvements and do not substantially change the perceived significance of the contribution. Overall, the paper was viewed as experimentally solid (especially after the rebuttal), but below the bar for ICLR in terms of novelty or impact.

**Reviewer Concerns:**

The rebuttal successfully addressed several scope-related concerns. In particular, the authors added experiments with additional PRMs, additional backbone models (including GPT-4o-mini), extensions to MCTS-style decoding, hyperparameter sensitivity analyses, and evaluations on harder benchmarks such as AMC23 and MinervaMath. These additions strengthen confidence that the observed volatility patterns are more generalizable than in the original manuscript. However, the core concerns raised by multiple reviewers remain. The proposed method is still largely a heuristic modification of existing guided decoding approaches, and the paper does not provide theoretical insight into why volatility scaling should be expected to work beyond empirical correlation. Moreover, even with the added experiments, performance gains are generally small on the main benchmarks and only become more pronounced in specific settings. As a result, while the rebuttal improves completeness and robustness, it does not materially change the assessment of the paper's overall contribution or impact.

**Reviewer Scores:**

- kJFH raised concerns primarily about limited generalization and modest performance gains. The rebuttal added experiments with additional PRMs and backbone models, which partially address these points. I expect the reviewer might have increased their score slightly (e.g., from 2 to 3), but likely would still view the overall contribution as insufficient for acceptance.
- EJpL expressed skepticism about the technical depth and significance of the contribution, and explicitly stated after the rebuttal that their concerns remained and that they intended to keep their initial assessment. I do not expect this reviewer's score would have changed with further discussion.
- m9QW emphasized the lack of theoretical grounding, reliance on limited model settings, and small empirical gains. While the rebuttal addressed generalization and experimental scope, the core concerns about contribution and impact remain. I would not expect a meaningful score change from this reviewer.
- ZvW3 viewed the paper as coherent and potentially acceptable but raised questions about robustness, hyperparameters, and practical significance. These concerns were largely addressed in the rebuttal. With a full discussion phase, this reviewer might have increased their score modestly (from 4 to 5), though this alone would not be sufficient to change the overall outcome.

---

### Decision · Program_Chairs · 2026-01-26

Reject